# Characterization and Neutral Atom Beam Surface Modification of a Clear Castable Polyurethane for Biomicrofluidic Applications

Atul Dhall [1,*], Tim Masiello [1], Suhasini Gattu [1], Matt Strohmayer [1], Logan Butt [1], Lewdeni Pathirannehelage Madhubhani Hemachandra [1], Sandra Schujman [1], Natalya Tokranova [1], Joseph Khoury [2], Satyavolu Papa Rao [1], Nathaniel Cady [1], Juan Andres Melendez [1] and James Castracane [1]

1    Colleges of Nanoscale Science and Engineering, SUNY Polytechnic Institute, Albany, NY 12203, USA;
     tmasiello@sunypoly.edu (T.M.); sgattu@sunypoly.edu (S.G.); mstrohmayer@sunypoly.edu (M.S.);
     lbutt@sunypoly.edu (L.B.); lhemachandra@sunypoly.edu (L.P.M.H.); sschujman@sunypoly.edu (S.S.);
     ntokranova@sunypoly.edu (N.T.); spaparao@sunypoly.edu (S.P.R.); ncady@sunypoly.edu (N.C.);
     jmelendez@sunypoly.edu (J.A.M.); jim.castracane@gmail.com (J.C.)
2    Exogenesis Corporation, Billerica, MA 01821, USA; jkhoury@exogenesis.us
*    Correspondence: adhall@sunypoly.edu; Tel.: +1-330-999-0750

**Abstract:** Polyurethanes (PU) are a broad class of polymers that offer good solvent compatibility and a wide range of properties that can be used to generate microfluidic layers. Here, we report the first characterization of a commercially available Shore 80D polyurethane (Ultraclear™ 480N) for biomicrofluidic applications. Studies included comparing optical clarity with Polydimethylsiloxane (PDMS) and using high-fidelity replica molding to produce solid PU structures from the millimeter to nanometer scales. Additionally, we report the first use of NanoAccel™ treatment in Accelerated Neutral Atom Beam (ANAB) mode to permanently roughen the surface of PU and improve the adhesion of breast cancer cells (MDA-MB-231) on PU. Surface energy measurements using Owens-Wendt equations indicate an increase in polar and total surface energy due to ANAB treatment. Fourier-transform infrared (FTIR) spectroscopy in attenuated total reflectance (ATR) mode was used to demonstrate that the treatment does not introduce any new types of functional groups on the surface of Ultraclear™ PU. Finally, applicability in rapid prototyping for biomicrofluidics was demonstrated by utilizing a 3D-printing-based replica molding strategy to create PU microfluidic layers. These layers were sealed to polystyrene (PS) bases to produce PU-PS microfluidic chips. Ultraclear™ PU can serve as a clear and castable alternative to PDMS in biomicrofluidic studies.

**Keywords:** Surface modification; contact angle; hard polyurethane; PDMS; neutral atom beam; surface energy; ATR-FTIR spectroscopy; microfluidics; cell viability; 3D printing

## 1. Introduction

Microfluidics involves the precise manipulation of fluids at submillimeter scales leveraging fabrication technologies developed by the semiconductor and microelectromechanical system (MEMS) industries [1]. A device built using microfluidic principles is commonly referred to as a micro total analysis system (μTAS) [2] or a Lab-on-a-Chip (LoC) [1]. There are certain distinct advantages to conducting experiments in a microfluidic setting rather than on the macroscale [1]. Microfluidic chips need substantially smaller sample volumes that reduce the cost of reagents. They allow simplified analysis of multiple samples in parallel to generate maximum data per batch, while also providing greater control of spatiotemporal fluid dynamics. Additionally, multiple targets can be analyzed on the

same sample. As these advantages translate well for use in biomedical research [3,4], a microfluidic setting provides an ideal platform for portable, point-of-care diagnostic devices [5].

Microfluidic systems almost always operate in the laminar flow regime leading to predictable fluid dynamics. In the absence of convective mixing, molecular transport is dominated by diffusion-based kinetics [1]. While these properties are desirable, they necessitate careful selection of materials to be used in the fabrication of microfluidic chips. For example, a material that strongly absorbs solutes can quickly deplete solutions in the channels of a microfluidic chip [6].

Polydimethylsiloxane (PDMS) is a silicon-based organic polymer that is most commonly used as a microfluidic material because of its optical clarity, biocompatibility, ease of molding, and flexibility [7]. However, PDMS has several disadvantages. It is susceptible to channel deformation due to its softness [8–10]. It is also known to leach uncured oligomers into the channel solution [11]. This can lead to additional steps to negate such leaching [12–14]. Hydrophobic interactions are a key driving force in biological phenomena and form the basis for drug design and discovery [15,16]. PDMS is known to strongly absorb small hydrophobic molecules [17,18]. Furthermore, the high vapor permeability of PDMS can lead to evaporation [19]. This can be harmful to cells at the scale of a microfluidic experiment [20,21]. Steps such as parylene coating [22] can resolve this issue but are not ideal for cell biology applications [1]. These drawbacks have resulted in a declining interest in PDMS biomicrofluidics [1,23]. Common alternatives to using PDMS include glass, polycarbonate (PC), polymethylmethacrylate (PMMA), cyclic olefin copolymers (COC), polyimides (PI), and polyurethanes (PU) [24,25].

PUs are a broad class of polymers that are most commonly formed by reacting a diisocyanate with a polyol [26]. Depending on the proprietary polyol curative used, the hardness of the resultant PU can vary from Shore A [6] through D. Unlike PDMS, PUs are compatible with organic solvents [25–28] and several aqueous solutions under 0.5 M [26]. This has led to many PU-based microfluidic studies [6,25–27,29–31]. Xia et al. [32] were the first to demonstrate that certain UV-curable PUs can be used in high-fidelity replica molding techniques past the micrometer scale with results similar to those for PDMS. Furthermore, the increased stiffness of PUs in comparison to PDMS allows fabrication of structures with greater aspect ratios that are less susceptible to ground and lateral collapse [33]. PUs have also been used to create thermally-actuated microfluidic valves that are more stable than PDMS valves due to decreased evaporation [30]. Finally, PUs have been used in biomicrofluidics, as soft bottom layers in hybrid microfluidic chips [31] and to form whole PU elastomeric chips for cell culture [6].

An increasing number of recent microfluidic studies involve chips made of hard plastics because of their suitability for use in modular microfluidics [34]. A similar trend can be seen in biomicrofluidic studies with a move away from PDMS in favor of hard plastics like PS that have traditionally played a large and well understood role in cell experiments in vitro [35]. Harder materials are also good candidates for surface modification techniques. One such technique—NanoAccel™ treatment in Accelerated Neutral Atom Beam (ANAB) mode [36,37]—can physically roughen surfaces using neutral argon atoms. It has been used on polyether ether ketone (PEEK) to improve its bioactivity towards cell attachment and proliferation.

Another trend in microfluidic fabrication is the increasing use of 3D-printing-based replica molding strategies [38,39] due to the low-cost of 3D printers, the reusability of 3D printed molds to create multiple microfluidic layers from the same mold and advancements in bonding rough layers to form sealed microfluidic chips.

This study stems from an ever-increasing body of research on hard plastics for microfluidics instead of PDMS due to its numerous disadvantages, the versatility and past success of softer PUs in microfluidics (including biomicrofluidics), and the relative ease with which PUs fit into simple 3D-printing-based replica molding and chip assembly strategies. To introduce a clear and castable alternative to PDMS in biomicrofluidic applications, we report the first characterization of a commercially available Shore D pour-and-cure-type, two-component PU resin (Ultraclear™ 480N

with hardness 80D from Hapco, Inc.). We also demonstrate control over its hydrophilic behavior, by describing the first utilization and evaluation of NanoAccel™ ANAB treatment on PU surfaces.

## 2. Materials and Methods

### 2.1. General Recipe for PDMS, Blue Silicone R-2374 and PU

PDMS (Sylgard® 184, Dow Silicones Corporation, Midland, MI, USA) and Blue Silicone R-2374 (Silpak, Inc., Pomona, CA, USA) had similar preparations—mixing 10:1 wt % (polymer:cross-linker). PU (Ultraclear™ 480N-10 and 480N-60, Hapco, Inc., Hanover, MA, USA) was prepared by mixing 1:1 wt % (part A:part B). Part A consists of 10–20 wt % of proprietary polyether polyol prepolymer capped with 80–90 wt % of 4,4′-methylene dicyclohexyl diisocyanate (H12MDI). Part B consists of 95–100 wt % of a proprietary polyether polyol combination. Ultraclear™ 480N-10 and 480N-60 have gel times of 10 and 60 min, respectively. After mixing, all uncured polymers were degassed for 30 min and cured at 65 °C for 2 h using a vacuum oven (Heraeus D-6450, Heraeus Instruments GmbH, Hanau, Germany).

### 2.2. Optical Transmittance Studies

UV-Vis-NIR transmission spectra from 200 nm–850 nm were measured for three polymers—PDMS (Sylgard® 184), fast-gelling PU (Ultraclear™ 480N-10), and slow-gelling PU (Ultraclear™ 480N-60). PDMS and PU were prepared, as described in the general recipe section. 1 mL of uncured samples was cured in 1.5 mL PS cuvettes (Brand GmbH + Co. KG, Wertheim, Germany) with a path length of 1 cm. A Cary® 50 Spectrophotometer (Agilent Technologies, Santa Clara, CA, USA) was used to collect spectral data. Three independent experiments were conducted in triplicate.

### 2.3. Feature Range Characterization

SU-8 (MicroChem Corp., Westborough, MA, USA) patterns on Si wafers were used to create masters. To maintain a rigid-flexible-rigid replica molding strategy, masters with features from the millimeter to the nanometer scale were used to develop flexible PDMS stamps. 1 cm layers of uncured PDMS were poured onto the masters enclosed in 150 mm (diameter) Petri dishes (Fisherbrand™, Thermo Fisher Scientific, Waltham, MA, USA) and cured. Cured PDMS stamps were peeled off the molds and used to generate PU replicas in a similar manner. Finally, PDMS layers were peeled off to leave PU blocks with the same features as the SU-8 patterned Si master.

### 2.4. Scanning Electron Microscopy

PU samples were cut into 2 cm wide squares for Scanning Electron Microscopy (SEM). PU samples were sputtered with Au/Pd (60:40) in a Denton Vacuum Desk IV® (Denton Vacuum, LLC, Moorestown, NJ, USA) using 30 mA for 75 s to avoid excessive charging, and then mounted with carbon tape. SEM images were collected using a LEO 1550 (Carl Zeiss Microscopy, LLC, Thornwood, NY, USA) at accelerating voltages between 1 kV–3 kV and magnifications between 125 X-25 kX.

### 2.5. Surface Modification by Corona Treatment

5 mm layers of PU were cured in 60 mm (diameter) Petri dishes. A BD-20AC Laboratory Corona Treater (Electro-Technic Products, Inc., Chicago, IL, USA) with a field effect electrode was used to treat each sample for 60 s.

### 2.6. Surface Modification by NanoAccel™ Treatment in ANAB Mode

5 mm layers of PU were cured in 60 mm (diameter) Petri dishes. Briefly, large clusters (~1000–5000 atoms/cluster) of argon gas were ionized and accelerated to 30 keV in a vacuum (with base pressure of 6.5E-7 Torr). By promoting cluster dissociation and deflecting charged cluster-fragments away, a beam of neutral argon atoms impinged on the PU surfaces with average kinetic energies in the

10–100 eV range. To ensure that all samples were subjected to the same vacuum and the parameter to be measured was affected by ANAB treatment alone, untreated samples were also placed in the NanoAccel™ tool with the beam blocked by a Ni mask.

### 2.7. Water Contact Angle Measurement

Water contact angle was measured for three different surfaces—Untreated PU, ANAB-treated PU, and corona-treated PU. 5 mm layers of PU were cured in 60 mm (diameter) Petri dishes. Cured samples were cut into 20 mm wide squares. A Cam-Plus Micro contact angle meter (ChemInstruments, Fairfield, OH, USA) was used to measure the water contact angle for 2 μL drops at ten randomly chosen spots across each surface. Drops were allowed to stabilize on the surface for 90 s before measurement of contact angle by the Half-Angle method. Three independent experiments were conducted with measurement of ten drops for each sample.

### 2.8. Atomic Force Microscopy

Images for ANAB-treated PU and untreated PU were taken using a Dimension Icon (Bruker, Billerica, MA, USA) in tapping mode for three randomly chosen spots on each sample. $10 \times 10~\mu m^2$ and $5 \times 5~\mu m^2$ images were captured and RMS roughness (Rq) values were recorded. 2D isotropic power spectral density plots were generated using NanoScope Analysis 1.8 (Bruker).

### 2.9. Surface Energy Estimation

Surface energy of untreated PU and ANAB-treated PU was estimated by the Owens-Wendt method [40–43] using three test liquids—water, formamide, and diiodomethane. Surface free energy parameters were taken from a PU study by Krol et al. and are listed in Table S1 [44]. Contact angle measurement was conducted as in Section 2.7. Average contact angle values were used to calculate the polar and dispersive parts of surface energy from the Owens-Wendt equations. Three independent experiments were conducted with measurement of ten drops for each sample.

### 2.10. Fourier-Transform Infrared Spectroscopy in Attenuated Total Reflectance Mode

Attentuated total reflectance-Fourier-transform Infrared (ATR-FTIR) spectroscopy was used for chemical surface characterization of untreated PU and ANAB-treated PU. A Tensor 27 (Bruker, Billerica, MA, USA) with a PIKE MIRacle™ ATR accessory (PIKE Technologies, Madison, WI, USA) and a ZnSe crystal was used to collect spectra between 520–4000 $cm^{-1}$ at a resolution of 4 $cm^{-1}$. Each spectrum collected was an average of 128 scans. PU samples were 2 mm thick and clamped down to the crystal using the accessory. Baseline correction was performed in SpectraGryph 1.2.

### 2.11. Cell Viability Studies

1 cm layers of PU were cured in 35 mm (diameter) Petri dishes. After UV sterilization for 1 h in a cell culture hood, samples were rinsed with 1X PBS (Phosphate Buffered Saline from Gibco™, Life Technologies, Thermo Fisher Scientific, Carlsbad, CA, USA) three times. 75000 MDA-MB-231 cells/mL were seeded onto PU surfaces with Corning™ DMEM (Dulbecco's Modified Eagle Media from Fisher Scientific, Hampton, NH, USA) + 10% FBS (Fetal Bovine Serum from Millipore Sigma, Burlington, MA, USA) + 1% Pen-Strep (Penicillin-Streptomycin from Millipore Sigma, Burlington, MA, USA) and grown in an incubator at 37 °C and 5% $CO_2$.

Live-dead staining by 0.4% Trypan Blue Dye (Bio-Rad, Hercules, CA, USA) exclusion was used to quantify the percentage of cell death after 24 h. Cells were monitored over the next 3 days and passaged to verify trypsinization on PU surfaces. Cells on sample sets were imaged before and after passaging to visualize differences in cell adhesion. Four runs were conducted in duplicate.

### 2.12. Chip Fabrication

A 3D-printing-based replica molding strategy was used to fabricate PU microfluidic layers. Autodesk® Inventor® designs were 3D printed using a Form 1+ (FormLabs, Somerville, MA, USA). All parts were printed at an axis resolution of 25 μm. Printed parts were separated from supporting frameworks and cleaned with isopropanol to remove uncured resin. Parts were then coated with 3 mL of Sigmacote® (Millipore Sigma, Burlington, MA, USA) to facilitate the removal of stamps. Uncured Blue Silicone R-2374 was poured into the printed parts and cured. Once cured, stamps were peeled off and used as molds for PU. Blue Silicone R-2374 stamps were preferred over PDMS stamps because they were found to be easier to remove from the 3D-printed masters. Uncured PU was poured into Blue Silicone R-2374 stamps. Once cured, stamps were peeled away to leave PU microfluidic layers. Holes were drilled for the inlets and outlets. PS cut-outs of required size were made from Falcon® cell culture Petri dishes (Corning Inc., Corning, NY, USA).

PU microfluidic layers were bonded to PS sheets using UV-curable adhesive NOA-63 (Norland Products, Inc., Cranbury, NJ, USA) to form PU-PS chips. Since chip features in biomicrofluidics are much larger than regular microfluidics, stamping-based methods [45,46] were unnecessary. A razor blade was used to level the adhesive on the microfluidic layer. The PS sheet was then gently placed on the NOA-63 coated-PU microfluidic layer and slight pressure was applied to remove excess adhesive and air bubbles while releasing boundary tension. Following a strategy used by Dang et al. [47], smaller PU-PS chips were designed with a sacrificial channel to prevent clogging of the microfluidic channels with NOA-63.

## 3. Results and Discussion

### 3.1. Optical Clarity

The optical clarity of a new microfluidic material is often compared to a traditional material like PDMS. As seen in Figure 1, the UV-Vis-NIR transmittance for both PU formulations (fast-gelling and slow-gelling) was similar to PDMS in the Vis-NIR range (380 nm–850 nm) and lower than PDMS in the UV range (200 nm–380 nm).

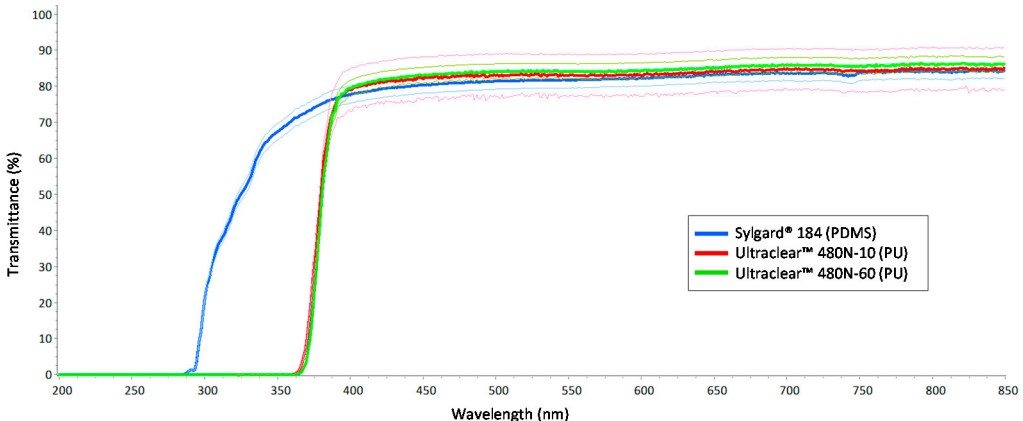

**Figure 1.** Comparison of the optical transmittance of PDMS and PU over the UV-Vis-NIR range. PU samples were prepared with two gel times (10 min for 480N-10 and 60 min for 480N-60). Both PU samples absorb in the UV range and transmit in the Vis-NIR range. The transmittance for all samples is similar above 380 nm. Thick lines represent averages of samples for each polymer. Thin lines above and below each thick line of the same color represent upper and lower first standard deviations for each polymer's dataset.

In general, PUs are known to yellow under prolonged UV irradiation [48]. The photodegradation mechanism for aromatic PUs is suspected to take place via a quinonoid route [49]. However,

manufacturers use proprietary additives to prevent UV degradation of PUs. Ultraclear™ PU is an example of a UV-resistant PU that absorbs completely in the UV range. PDMS is also known to degrade under UV radiation without protective additives [50,51].

The spectra in Figure 1 are consistent with softer PU formulations (shore A), as characterized by Domansky et al. [6]. Ultraclear™ PU's similarity in optical transmittance with PDMS at higher wavelengths is useful as these wavelengths are relevant for most biological assays. PU 480N-10 samples gelled quickly and needed immediate degassing under a strong vacuum. This resulted in slight inconsistencies while preparing such samples. Since the optical properties of both PU formulations were identical, PU 480N-60 was chosen for all further experiments to ease sample preparation.

### 3.2. Microfluidic Range Achievable

Figure 2 illustrates the replica molding strategy used to fabricate solid Ultraclear™ PU structures. Rigid-flexible-rigid steps allowed easy separation of layers. Xia et al. [32] have previously demonstrated high-fidelity replication of Au masters to UV-curable PU replicas using intermediate PDMS stamps. Ultraclear™ PU performs similarly in high-fidelity replica molding.

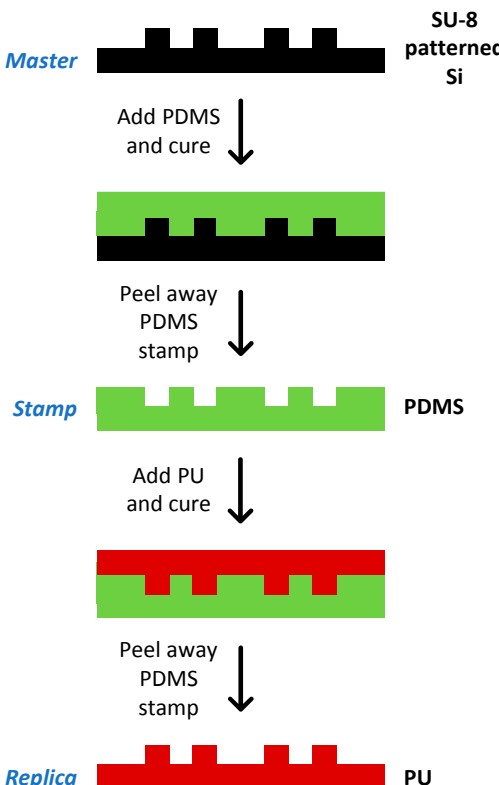

**Figure 2.** Schematic of replica molding strategy used to create Ultraclear™ PU structures from the millimeter to the nanometer scale. A rigid (SU-8 patterned Si)-flexible (PDMS)-rigid (PU) molding strategy was used to allow easy separation of layers.

SEM images in Figure 3 depict identical replication of Si-based masters using intermediate PDMS stamps in the 130 nm–1.5 mm range. This demonstrates that Ultraclear™ PU can be used to make structures across the entire breadth of feature sizes used in microfluidics and can serve as an alternative to PDMS. Additionally, unlike PDMS, PU microfluidic channels are not prone to sagging [9,52]. Cell-based microfluidic chips have much larger feature sizes than chemistry-based microfluidic chips. In fact, microfluidic channels in cell-based chips can often be up to 1 mm wide [52,53]. A wide variety of shapes were chosen for characterizing the feature range achievable to encompass large

microfluidic reservoirs and channels (Figure 3A,B) and smaller features to help with sorting and alignment (Figure 3C,F).

Zhang et al. [33] have previously demonstrated that UV-curable PUs, being stiffer than PDMS, allow fabrication of structures with aspect ratios up to 12 without being susceptible to ground and lateral collapse. While a similar trend is expected for Ultraclear™ PU, its applicability in cell-based microfluidics with respect to feature sizes is adequately depicted by the moderate aspect ratio features in Figure 3.

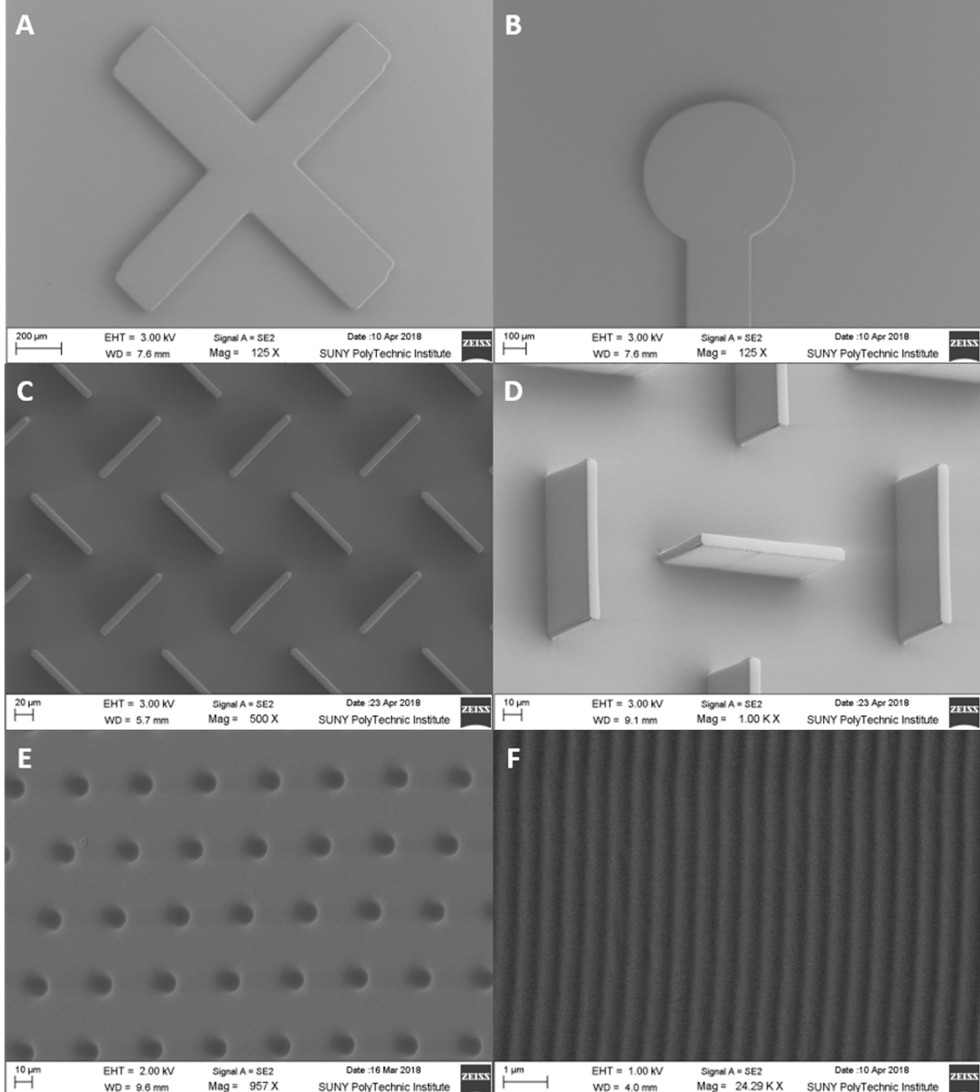

**Figure 3.** SEM images of PU structures demonstrating high-fidelity replication of features from the millimeter scale to the nanometer scale. A rigid (SU-8 patterned Si)-flexible (PDMS)-rigid (PU) replica molding strategy was used to create PU replicas of features, originally etched in SU-8 patterned Si wafers, with PDMS as a replication intermediate. Images are ordered by decreasing feature size from panel A through F. While panels A and B depict inverses of microfluidic reservoirs and channels, panels C through F depict typical features for sorting and alignment in microfluidic chips. Fidelity of the replica molding procedure was high as features in PU were identical in size to those in the SU-patterned Si wafers. (**A**) A 1.5 × 0.2 mm cross. Scale bar: 200 μm. (**B**) Inverse of a microfluidic channel with a 0.5 mm inlet (diameter) and 0.25 mm wide channel. Scale bar: 100 μm. (**C**) A staggered array of perpendicular bars (80 × 5 × 20 μm). Scale bar: 20 μm. (**D**) Zoomed in image of the bars in (**C**). Scale bar: 10 μm. (**E**) An inline array of 10 μm wells (diameter). Scale bar: 10 μm. (**F**) 130 nm wide parallel lines. Scale bar: 1 μm.

### 3.3. Hydrophilic Surface Modification

Unlike shore A PUs [6], Ultraclear™ PU was found to be hydrophobic after curing. To demonstrate applicability in cell-based microfluidics, we utilized two methods to make Ultraclear™ PU hydrophilic–corona treatment and ANAB treatment. The water contact angle for untreated PU samples was 106°. Figure S1 depicts the temporary hydrophilic gain and eventual hydrophobic recovery of PU samples after corona treatment. The contact angle decreased to 34.6° immediately after treatment. As expected, the samples experienced hydrophobic recovery and the contact angle rose back to 100.4° after 24 h.

While a temporary reduction of the contact angle can be useful for bonding steps in chip assembly, permanent reduction is desirable for cell adhesion and growth. Since cell adhesion is favored on surfaces with moderate hydrophilicity [52,54], permanent surface modification was needed to make Ultraclear™ PU suitable for biomicrofluidic applications. NanoAccel™ treatment [36,37] in ANAB mode was used to permanently modify the surface.

Figure 4A depicts a permanent increase in hydrophilicity after ANAB treatment (samples tested up to six months later). An increase in beam flux gradually decreased the water contact angle. Figure 4B,C are representative AFM images for untreated and ANAB-treated PU samples, respectively. The average Rq value for surface roughness increased from 2.03 nm for untreated PU samples to 12.5 nm for ANAB-treated samples. This corroborates the data shown in Figure 4A as rougher surfaces (on the scale of ANAB treatment results) tend to have lower water contact angles. For the representative images shown in Figure 4B,C, 2D isotropic power spectral density plots were generated, as shown in Figure S2.

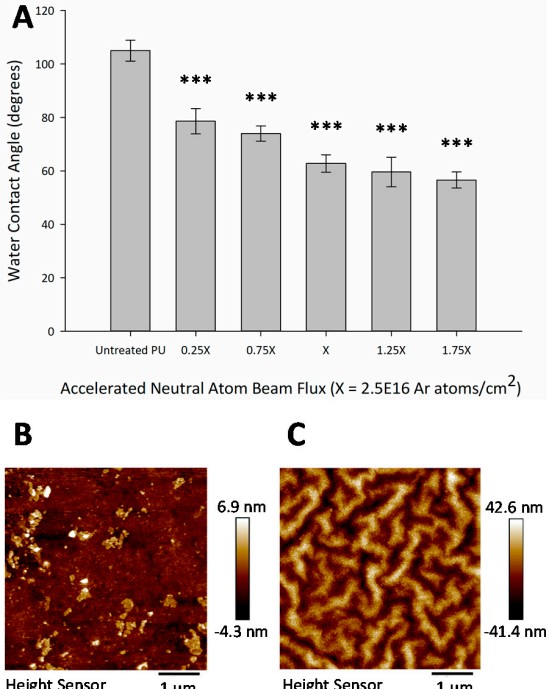

**Figure 4.** NanoAccel™ treatment using ANAB mode makes Ultraclear™ PU surfaces hydrophilic. A neutral Ar atom beam with average energy in the 10–100 eV range impinged on the PU surfaces. (**A**) A range of values for atomic flux were used to identify an optimal beam flux. Values are expressed in terms of this optimal flux, $X = 2.5 \times 10^{16}$ atoms/cm$^2$. Water contact angles were measured at 10 randomly chosen spots on each sample. Data represent means. Error bars are standard deviations. One-way ANOVA: $p < 0.001$. Post hoc: *** represents $p < 0.001$ in comparison to the untreated control using Dunnett's method. (**B**) Untreated PU has an average roughness of 2.03 nm. Scale bar: 1 μm. (**C**) ANAB-treated PU has a pattern of rough features with average roughness of 12.5 nm. Scale bar: 1 μm.

The Owens-Wendt method is commonly used to estimate solid surface energy for polymers by assuming it to be made up of two interactions—polar and dispersive. While the dispersive component accounts for van der Waals and non-site specific interactions, the polar component accounts for dipole-dipole, hydrogen bonding, and site specific interactions [42].

To confirm an increase in hydrophilic character as shown by water contact angle data (Figure 4A), we estimated the surface energy of Ultraclear™ PU using the Owens-Wendt equations. Table 1 contains values for surface energy calculated from contact angle data for three liquids (water, formamide and diiodomethane). As shown, the polar component of surface energy for PU samples increased with treatment flux. However, the dispersive component remained largely unchanged. Thus, the total surface energy of the PU samples increased. This indicates an increase in hydrophilic character. Testing liquids were chosen to include a range of polar surface free energy parameters (water: High; formamide: Moderate; and diiodomethane: Low). As expected, contact angles for treated samples decreased sharply for water, moderately for formamide and gradually for diiodomethane. Similar to the trend in data from Figure 4A, the effect of ANAB treatment on surface energy of Ultraclear™ PU tends to flatten out beyond an optimal flux value of $2.5 \times 10^{16}$ atoms/cm$^2$ (X).

**Table 1.** Surface energy values calculated by the Owens-Wendt method.

| PU Treatment | Surface Energy (mN/m) | | |
|:---:|:---:|:---:|:---:|
| | Polar | Dispersive | Total |
| Untreated | 0.1 | 26.7 | 26.8 |
| 0.25X | 7.9 | 25.9 | 33.8 |
| 1X | 12.6 | 24.9 | 37.5 |
| 1.75X | 14.9 | 25.0 | 39.9 |

Neutral Atom Beam Flux, X = $2.5 \times 10^{16}$ atoms/cm$^2$.

Finally, to investigate chemical bonding group changes due to ANAB treatment, ATR-FTIR spectra of Ultraclear™ PU samples were collected. The stacked spectra in Figure 5A clearly demonstrate that ANAB treatment did not introduce any new types of functional groups on the PU surfaces. Peaks in the 1700 cm$^{-1}$ region, 2900 cm$^{-1}$ region and 3400 cm$^{-1}$ region were identified as C=O, CH$_2$ and NH groups, respectively [55,56]. The peak at 2270 cm$^{-1}$ was identified as N=C=O from the H12MDI monomer [55,57]. While all the spectra are similar in Figure 5A, the NH group influence gets slightly stronger with increasing treatment flux. Figure 5B shows the NH group peaks in the 3180–3590 cm$^{-1}$ range and the uptick in their absorbance. NH groups directly affect the availability for polar interactions.

Qualitatively, ruling out the presence of new types of functional groups due to ANAB treatment of the PU surface is straightforward. However, we must note some concerns with quantifying NH influence from the ATR-FTIR spectra. ATR-FTIR data are dependent, among other parameters, on the smoothness of the samples used, the pressure applied by the clamp to force the sample against the crystal and the aggressiveness of postprocessing spectra with baseline correction. Despite these limitations, we notice an uptick in NH peaks with treatment flux. At the same time, surface energy measurements using two-parameter (polar and dispersive) models like the Owens-Wendt equations are unavoidably dependent on the accuracy of the contact angles measured (manufacturer reported accuracy of 0.8°) and the surface free energy parameters chosen for the testing liquids. Thus, the trends in our data from surface energy calculations and ATR-FTIR spectra coupled with the increase in surface roughness from Figure 4B,C seem to agree even though they are collected from independent methods.

Unlike other methods that introduce OH group influence on the surface and add to the polar and total surface energy of PU samples [40], NanoAccel™ treatment of PU adds to the polar component of surface energy by increasing NH group influence on the surface. We attribute this to an increased roughness due to ANAB treatment. This leads to increased surface area and consequently more NH

groups on the surface available for polar interactions. The overall effect is an increase in polar surface energy (and thus, total surface energy).

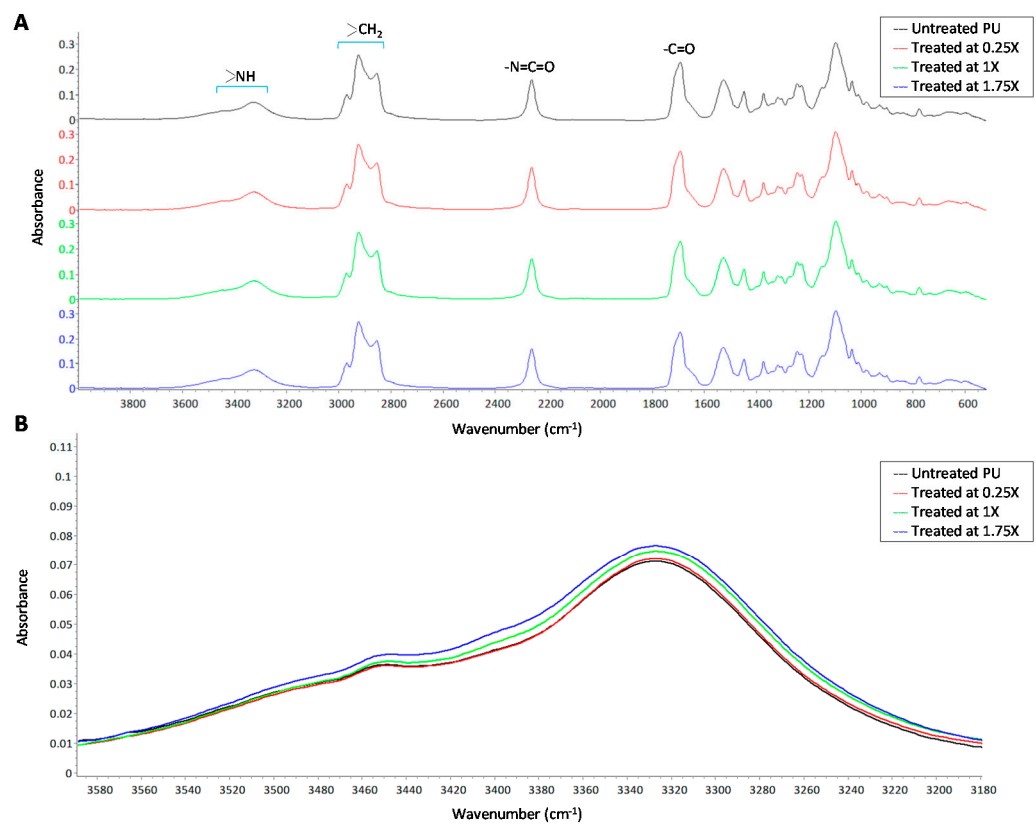

**Figure 5.** ATR-FTIR spectra for ANAB-treated Ultraclear™ PU surfaces. (**A**) Stacked spectra for samples exposed to a range of atomic flux, X = $2.5 \times 10^{16}$ atoms/cm$^2$. Peaks in the 1700 cm$^{-1}$ region, 2270 region, 2900 cm$^{-1}$ region and 3400 cm$^{-1}$ regions represent C=O, N=C=O, CH$_2$ and NH groups, respectively. (**B**) Zoomed in spectral overlap of wavenumbers 3180–3590 cm$^{-1}$ indicating a mild increase in NH group influence with increasing treatment flux.

### 3.4. Cell Viability

To demonstrate the applicability of Ultraclear™ PU in biomicrofluidics, cell viability was assessed. MDA-MB-231 cells were used because of their adhesive behavior and abundant use in microfluidics [58–61]. The contact angle for polystyrene used in tissue culture ranges between 55.8° to 63.5° [62]. To tailor Ultraclear™ PU's surface for moderate hydrophilicity [54] and optimal cell adhesion, a beam flux of $2.5 \times 10^{16}$ atoms/cm$^2$ (X) was chosen to analyze cell viability by measuring percentage cell death using Trypan Blue staining. As shown in Figure 6, ANAB-treated samples displayed significantly reduced cell death after 24 h when compared to untreated PU. MDA-MB-231 cell morphology was also consistent with morphology when grown on traditionally used polystyrene.

Upon reaching confluence, cells were passaged using trypsin. Figure S3 demonstrates the persistence of a similar trend in cell adhesion after passaging, indicating that ANAB treatment led to a sustained improvement in cell adhesion properties of Ultraclear™ PU.

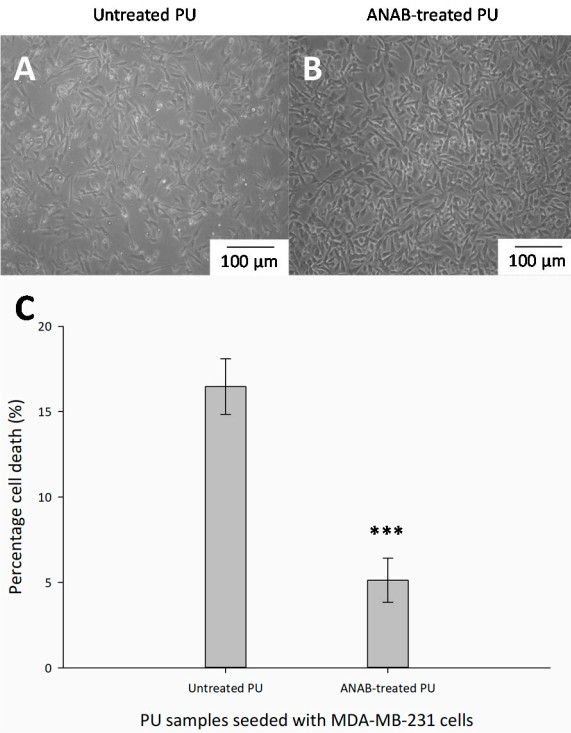

**Figure 6.** NanoAccel™ treatment using ANAB mode promotes cell adhesion on Ultraclear™ PU surfaces. 75000 MDA-MB-231 cells were seeded onto 5 mm thick PU cured in 35 mm (diameter) Petri dishes and imaged after 24 h. (**A**) Cells on untreated PU. Scale bar: 100 μm. (**B**) Cells on ANAB-treated PU. Scale bar: 100 μm. (**C**) Percentage cell death after 24 h was quantified using Trypan Blue staining for both treated and untreated PU. Percentage cell death was significantly lower for treated PU. Data represent means. Error bars are standard deviations. *** represents $p < 0.001$ for a two-tailed t-test.

*3.5. Chip Fabrication*

3D-printing-based replica molding strategies are useful for rapid prototyping and simple production of microfluidic layers. However, the prints generated are rough, and when compared to traditional microfluidics, necessitate the use of adhesive-assisted bonding for reliable sealing.

As shown in Figure 7, we used a simple 3D-printing-based replica molding strategy to fabricate PU microfluidic layers. Microfluidic channels in traditional cell-based chips can often be up to 1 mm wide [52,53]. With the Form 1+ 3D printer, we were able to reliably print microfluidic features well under this range with the smallest channel widths down to 250 μm. Taking advantage of the relatively large feature sizes, we were able to avoid stamping methods for adhesive bonding [45,46].

To demonstrate the applicability of Ultraclear™ PU across cell-based microfluidics, we used PS as the bottom layer for our chips bonded to a microfluidic PU layer using NOA-63. As shown in Figure 8, two versions of PU-PS chips were fabricated to showcase the rapid prototyping potential of this method—a smaller chip with a single reservoir encompassed by a sacrificial channel and a larger chip with three interconnected reservoirs. A sacrificial channel [47] was used on the smaller chip to prevent clogging the microfluidic channels with the adhesive. The larger chip did not need a sacrificial channel.

Overall, our chip fabrication method completely avoids the disadvantages of PDMS by using PU, eliminates the use of stamping methods during assembly, eliminates the need to use photolithography by keeping feature sizes attainable via 3D-printing-based replica molding, and incorporates the most logical material for cell-based microfluidics (PS) as a bottom layer.

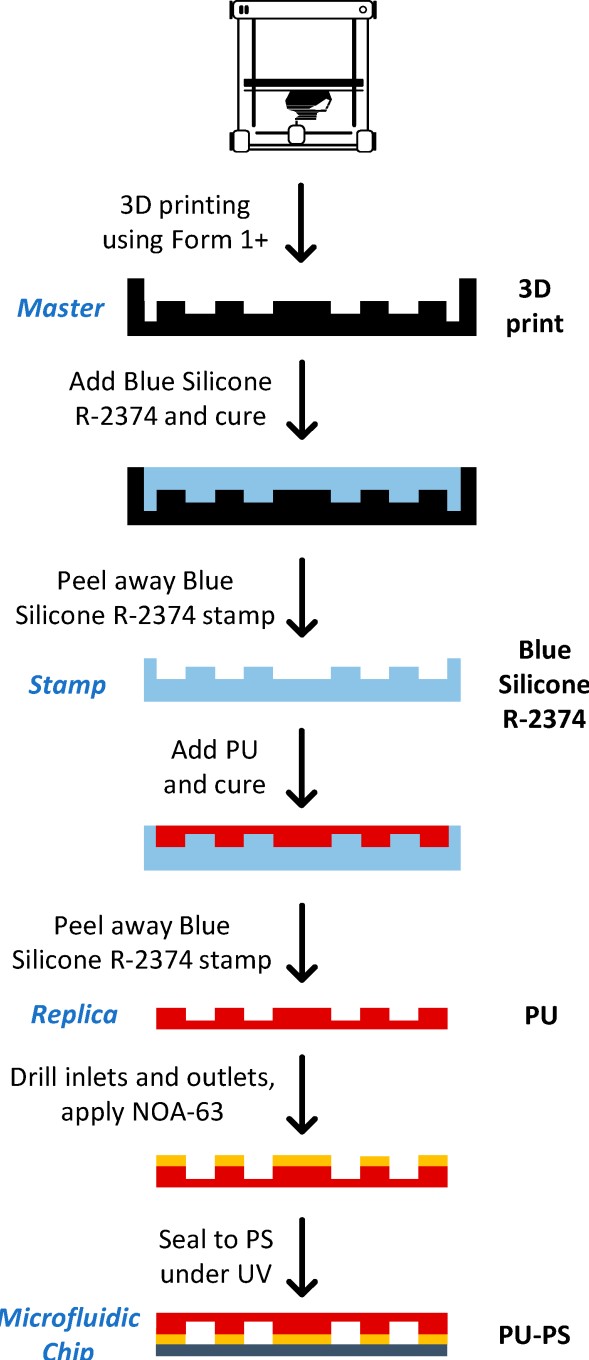

**Figure 7.** Schematic of 3D-printing-based replica molding strategy used to create PU-PS microfluidic chips. A rigid (3D print)-flexible (Blue Silicone R-2374)-rigid (PU) molding strategy was used to allow easy separation of layers. PU microfluidic layers were then sealed to PS using UV-curable NOA-63. 3D printer icon credit: Bryan Allen from The Noun Project (https://thenounproject.com/term/3d-printer/41229/).

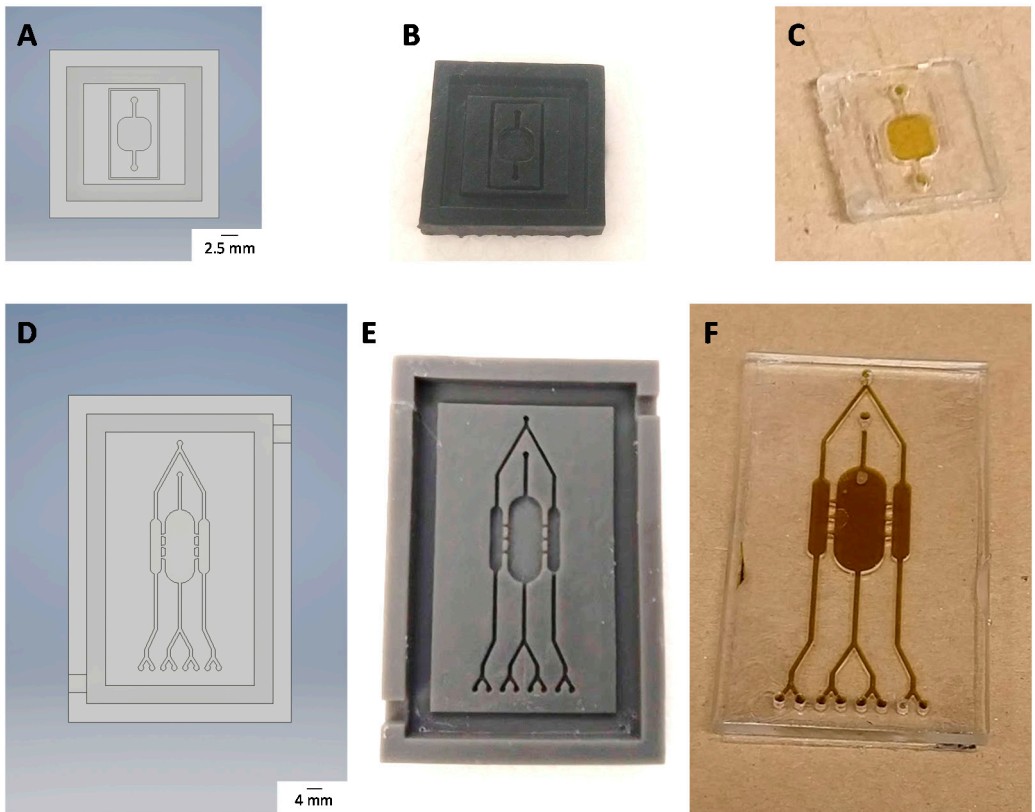

**Figure 8.** Fabrication of PU-PS microfluidic chips using the 3D-printing-based replica molding strategy and adhesive bonding. Autodesk® Inventor® was used to 3D print mold designs (panels A and D). Once printed (panels B and E), molds were used to create PU-PS microfluidic chips (panels C and F) according to the replica molding schematic shown in Figure 6. (**A–C**) Fabrication of a chip with small features using a sacrificial channel to prevent clogging by adhesive. (**D–F**) Fabrication of a chip with larger features where sacrificial channels are not needed. (**A**) Inventor® diagram of mold for microfluidic PU layer of smaller PU-PS chip. Mold dimensions are 25 × 25 × 5 mm. Chip consists of a single chamber with one inlet and one outlet (0.5 mm radius). The channel is 0.5 mm wide and 1 mm deep. The chamber is a 5 × 5 mm rounded square with a fillet radius of 1.5 mm. The sacrificial channel is 0.25 mm wide and 1 mm deep. Scale bar: 2.5 mm. (**B**) 3D print of panel A in black resin (meant for models with intricate details). (**C**) A 15 × 15 mm sealed PU-PS microfluidic chip filled with food dye. The sacrificial channel is partially clogged with adhesive leaving the chamber and microfluidic channel adhesive-free. (**D**) Inventor® diagram of mold for microfluidic PU layer of larger PU-PS chip. Mold dimensions are 88 × 60 × 7 mm. Chip consists of a wide central chamber with two narrow outer chambers on each side. The central chamber is a 20 × 8 mm rounded rectangle with a fillet radius of 4 mm. The outer chambers are 15 × 3 mm rounded rectangles with fillet radii of 1.5 mm. Chambers are interconnected by channels tapering towards the outer chambers (1 mm to 0.5 mm). Channels are 1 mm wide and 1.5 mm deep. Chip has 8 inlets (lower) and 2 outlets (upper) with radii of 0.75 mm. Scale bar: 4 mm. (**E**) 3D print of panel D in grey resin (meant for general purpose prototyping). (**F**) A 70 × 40 mm sealed PU-PS microfluidic chip filled with food dye. The larger feature size allows sacrificial channel-free fabrication.

## 4. Conclusions

A declining interest in using PDMS for microfluidics has led to the characterization of novel materials for rapid prototyping. This trend has continued in biomicrofluidic research. PU-based polymers are often used in biomicrofluidics because they do not absorb hydrophobic molecules and have better solvent compatibility and stiffness than PDMS. We report the first characterization of a commercially available Shore 80D PU for biomicrofluidic applications.

Ultraclear™ PU has an optical transmittance similar to PDMS in the Vis-NIR range. It can be used reliably, with replica molding strategies, to fabricate solid structures across the breadth of the microfluidic range. Unlike other PUs, Ultraclear™ PU is hydrophobic after curing. Corona treatment causes a temporary gain in hydrophilicity. To demonstrate applicability for biomicrofluidic studies, we report the first use of NanoAccel™ neutral atom beam surface modification of PU surfaces to permanently roughen the surface of Ultraclear™ PU and reduce its water contact angle. Surface energy measurements using Owens-Wendt equations demonstrate an increase in polar surface energy with increasing treatment flux. ATR-FTIR spectra prove that no new functional groups are introduced on the surface due to treatment. The improved surface roughness and hydrophilic behavior also favors MDA-MB-231 cell adhesion. Lastly, to demonstrate applicability in rapid prototyping, a 3D-printing-based replica molding strategy is utilized to create PU microfluidic layers that are sealed to PS using adhesive bonding. As a proof of concept, two versions of PU-PS chips were made. Overall, we demonstrate that Ultraclear™ PU is a clear, castable alternative to PDMS for use in rapid prototyping and biomicrofluidics. Future directions include testing the potential of NanoAccel™ treatment to pattern Ultraclear™ PU surfaces with specific hydrophilic and hydrophobic regions and incorporating such strategies into microfluidic chip usage.

**Supplementary Materials:** The following are available online at http://www.mdpi.com/2571-9637/2/1/9/s1. Figure S1: Corona treatment of Ultraclear™ PU results in a temporary gain of hydrophilicity and eventual hydrophobic recovery, Figure S2: Power spectral density analysis for representative AFM images of Ultraclear™ PU with and without NanoAccel™ treatment in ANAB mode, Table S1: Parameters used to estimate surface energy by the Owens-Wendt method, Figure S3: Improved cell adhesion of ANAB-treated PU persists even after passaging cells.

**Author Contributions:** Standard CRediT-based descriptions are as follows. Conceptualization: A.D., M.S., L.B., N.T. and S.P.R.; Data Curation: A.D.; Formal Analysis: A.D.; Funding acquisition: J.A.M. and J.C.; Investigation: A.D., T.M., S.G., L.P.M.H. and S.S.; Methodology: A.D., M.S., L.B., L.P.M.H., N.T., J.K. and S.P.R.; Project Administration: N.T., J.A.M. and J.C.; Resources: J.K., N.C. and J.C.; Supervision: J.A.M. and J.C.; Validation: A.D., L.P.M.H., N.T. and S.P.R.; Visualization: A.D. and N.C.; Writing–Original Draft: A.D.; Writing–Review & Editing: A.D., T.M., L.B., L.P.M.H., N.T. and N.C. Specifically, A.D. conducted all the experiments and wrote the manuscript. T.M. and L.P.M.H. helped prepare samples and collect preliminary data for the cell viability experiments (Figure 6 and Figure S3). S.G. helped image the samples used to identify the feature range achievable (Figure 3). M.S., L.B. and L.P.M.H. helped develop the 3D-printing-based replica molding strategy and an initial chip design (Figures 7 and 8D). S.S. helped image samples to corroborate surface roughness data (Figure 4B,C). N.T., J.K. and S.P.R. helped design experiments to increase PU hydrophilicity using NanoAccel™ treatment in ANAB mode (Figure 4A). N.C. and N.T. helped structure the manuscript. J.A.M. and J.C. supervised the project.

**Funding:** This research was funded by SUNY Polytechnic Institute, Albany, NY.

**Acknowledgments:** We would like to thank: Bon Phok from Exogenesis Corporation (Billerica, MA) for processing samples on the NanoAccel™ tool, Mike Walsh (Exogenesis Corporation) for discussions on process conditions and transporting samples between Billerica and Albany, and Michael Murphy (SUNY Polytechnic Institute, Albany, NY) for initial guidance with using the ATR-FTIR tool.

**Conflicts of Interest:** J.K. is an employee and owns shares of Exogenesis Corporation (Billerica, MA). All other authors declare no conflict of interest. The funders had no role in the design of the study; in the collection, analyses, or interpretation of data; in the writing of the manuscript, or in the decision to publish the results.

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
