# Peer review of "Characterization and Neutral Atom Beam Surface Modification of a Clear Castable Polyurethane for Biomicrofluidic Applications"

_surfaces, doi:10.3390/surfaces2010009_

Round 1
Reviewer 1 Report
The paper is rather well written. All experimental details are clearly
explained. The findings may be usefull for the readership. However, the
article is rather long, but the actual problem, which is the surfae
modification of PU by argon atoms, is hardly adressed. For a good paper,
this should be extended, e.g. by more extensive (chemical) surface characterisation, not only the contact angle. Nevertheless, it seem to work, and the article could be published.
Author Response
PDF response is attached

Reviewer 2 Report
The work entitled "Characterization and neutral atom beam surface modification of a clear castable polyurethane for biomicrofluidic applications" described a first characterization of a commercially available Shore 80D 24 polyurethane for biomicrofluidic applications.
Generally, the work is well done and the information is scientifically sound. There is a clear lack of discussion of the results, that should be addressed (see the indications below). Still, there is a lot of merit in the research and the subject is quite interesting.
These are some details that should be revised before publication:
1. The first five lines of the abstract are just introductory material. Here, it is not necessary to give such an extensive background. Please reduce the information and perhaps replace it with more data from the results/conclusions. At the moment the abstract is not attractive for the readers.
2. Please confirm if the number of key words is correct (they are too many).
3. Materials and methods:
- (A) by weight or weight %, should be replaced by wt%
- (D) SEM testing: please provide the electron beam intensity (accelerating voltage) and magnification. Also, it would be important to add the current used during sputtering.
- (E/F) "5 mm layers of PU were cured in 60 mm petri dishes" this referrers to each dimension, diameter? Please add the details.
- (G) the size of the drops should be specified.
- (I) please provide the complete terminology for "10% FBS + 1% Pen-Strep".
4. The title of the section Results, should be altered to "Results and Discussion" since the authors have done some discussion together with the presentation of the results. However, it would be important to deepen this discussion a little further, particularly in the cell viability (D) sub-section. Also, it would be important to point the advantages of this chip fabrication method in light of conventional technologies and materials.
5. The last section cannot be called a "Discussion", since the authors basically did a summary of the acquired data. This is more a conclusion section than an actual discussion. I would recommend the authors to combine "Results and Discussion" in the same section and explore a little further the details of their results.
Finally, there were some small spelling mistakes that should be corrected after another careful reading of the manuscript.
Author Response
PDF response is attached

Reviewer 3 Report
The papers reports novel approach, which will be interesting for a wide audience. However, some improvements are still required.
1. In Fig. 1 the authors presented average values of transmittance. Please also present standard deviations if possible to reveal any consistency of the results obtained.
2. The authors presented the results of surface wettability. Did the authors consider measurements of the surface energy to prove its hydrophilic character?
3. Did the authors observe the range of surface roughnesses, which result in significant surface wetting behavior changes after neutral atom beam surface modification.
4. Can the approach of surface treatment be used for surface processing of other polymers, which are different from PU?
Author Response
PDF response is attached

Round 2
Reviewer 2 Report
The manuscript is significantly improved. The authors did a good job addressing all the reviewers comments. The manuscript is now worthy of publication.